# Associations between Sex and Risk Factors for Predicting Chronic Kidney Disease

**DOI:** 10.3390/ijerph19031219

**Published:** 2022-01-22

**Authors:** Hao-Yun Kao, Chi-Chang Chang, Chin-Fang Chang, Ying-Chen Chen, Chalong Cheewakriangkrai, Ya-Ling Tu

**Affiliations:** 1Department of Healthcare Administration and Medical Informatics, College of Health Sciences, Kaohsiung Medical University, Kaohsiung 80708, Taiwan; haoyun@kmu.edu.tw; 2School of Medical Informatics, Chung Shan Medical University & IT Office, Chung Shan Medical University Hospital, Taichung City 40201, Taiwan; amy0988147957@gmail.com; 3Department of Information Management, Ming Chuan University, Taoyuan City 33300, Taiwan; 4Department of Otorhinolaryngology, Head and Neck Surgery, Jen-Ai Hospital, Taichung City 41222, Taiwan; 5Cancer Medicine Center, Jen-Ai Hospital, Taichung City 41222, Taiwan; 6Basic Medical Education Center, Central Taiwan University of Science and Technology, Taichung City 40601, Taiwan; 7Department of Medical Education and Research, Jen-Ai Hospital, Taichung City 41222, Taiwan; 8Department of Obstetrics and Gynecology, Division of Gynecologic Oncology, Faculty of Medicine, Chiang Mai University, Chiang Mai 50200, Thailand; chalong.c@cmu.ac.th; 9Center for General Education, National Taichung University of Science and Technology, Taichung City 40401, Taiwan; a54sandra@gmail.com

**Keywords:** gender differences, chronic kidney disease, machine learning, risk factors

## Abstract

Gender is an important risk factor in predicting chronic kidney disease (CKD); however, it is under-researched. The purpose of this study was to examine whether gender differences affect the risk factors of early CKD prediction. This study used data from 19,270 adult health screenings, including 5101 with CKD, to screen for 11 independent variables selected as risk factors and to test for the significant effects of statistical Chi-square test variables, using seven machine learning techniques to train the predictive models. Performance indicators included classification accuracy, sensitivity, specificity, and precision. Unbalanced category issues were addressed using three extraction methods: manual sampling, the synthetic minority oversampling technique, and SpreadSubsample. The Chi-square test revealed statistically significant results (*p* < 0.001) for gender, age, red blood cell count in urine, urine protein (PRO) content, and the PRO-to-urinary creatinine ratio. In terms of classifier prediction performance, the manual extraction method, logistic regression, exhibited the highest average prediction accuracy rate (0.8053) for men, whereas the manual extraction method, linear discriminant analysis, demonstrated the highest average prediction accuracy rate (0.8485) for women. The clinical features of a normal or abnormal PRO-to-urinary creatinine ratio indicated that PRO ratio, age, and urine red blood cell count are the most important risk factors with which to predict CKD in both genders. As a result, this study proposes a prediction model with acceptable prediction accuracy. The model supports doctors in diagnosis and treatment and achieves the goal of early detection and treatment. Based on the evidence-based medicine, machine learning methods are used to develop predictive model in this study. The model has proven to support the prediction of early clinical risk of CKD as much as possible to improve the efficacy and quality of clinical decision making.

## 1. Introduction

Chronic kidney disease (CKD) is a global public health problem that is related to severe morbidity, mortality, and high medical resource utilization [1]. In 2017, the global estimated prevalence of CKD was 9.1% with a total of 69.75 million cases and 1.2 million deaths. [2]. The global estimated prevalence of CKD is 9.1%. According to Taiwan’s Ministry of Health and Welfare, CKD accounts for the largest proportion of health insurance claims in Taiwan, with a total of 364,000 patients accounting for costs of approximately TWD 51.3 billion in 2018 [3]. Population aging and the associated increase in hypertension continue to raise the prevalence of hyperlipidemia and hyperglycemia and thus the incidence of CKD. Generally, two basic definitions of CKD are used: 1. Kidney damage for >3 months, including structural and functional abnormalities, which may be pathological, blood, urine, or imaging abnormalities; and 2. Glomerular filtration rate (GFR) < 60 mL/min/1.73 m^2^ for over 3 months. CKD is usually divided into five stages based on GFR [4] (see Table 1). However, the adverse consequences and clinical risks of renal function insufficiency in early cases differ between genders.

The current literature mostly focuses on differences in kidney disease prognoses for men and women of average age. In men, urinary albumin excretion, plasma glucose, and systolic blood pressure are the most important predictors of severely declined renal function [5,6]. In addition, waist circumference and the cholesterol-to-high-density lipoprotein ratio are positively associated with the maintenance of kidney function. Relative to women, plasma glucose and systolic blood pressure are risk predictors of decreased renal function, and triglycerides are positively related to maintaining renal function [5,6]. Related epidemiological studies show that the incidence of CKD in women is lower than in men [7,8]. In addition, estrogen was shown to protect kidney function in related animal models, including glomerular sclerosis reduction and ischemic damage prevention [9,10]. Ricardo et al. (2019) [11] found a significant difference in eGFR decline between men and women (−1.09 mL/min/1.73 m^2^ for women and −1.43 mL/min/1.73 m^2^ for men (*p* < 0.001)).

However, only few researchers focused on gender differences among adults [12]. Current screening standards provide measurements, but it is difficult to effectively predict CKD precisely [13]. At present, at least 2 million patients are diagnosed with CKD in Taiwan, but only 3.5% can secure an early diagnosis for timely treatment, leading to the loss of 25% of kidney function before diagnosis. Thus, early detection helps to effectively prevent deterioration. Evidence-based medicine requires gender heterogeneity to be taken into account in CKD deterioration to inform the risk assessment, monitoring, and prognosis. A recent survey showed that Taiwan has a high incidence of CKD, but public awareness of the condition is extremely low. According to the 2012 report of the United States Preventive Services Task Force and the American College of Physicians, kidney function is insufficient for asymptomatic individuals, and the effective tools for CKD detection are lacking [14]. Regardless of the risk factors, the American Society of Nephrology strongly recommends regular CKD detection [14].

This study used preventive health screening data from Taiwanese adults to assess the impact of gender differences as risk factors for CKD in order to achieve early prediction of reduced renal function. Since 2012, Taiwan implemented a 5-year plan for CKD prevention and quality care improvement to reduce the need for dialysis after kidney transplantation and improve the 5-year survival rate. However, in 2017, Taiwan reported 275,000 cases of CKD, and 6743 of these resulted in deaths. The uncertainty of diagnosis usually results from heterogeneity screening and clinical practices; thus, an accurate tool is needed for early prediction to ensure that potential patients receive and comply with preventive health check-ups. Data mining has been successfully used to build predictive models for healthcare prediction tasks [15,16,17,18,19,20,21]. The present study sought to evaluate the novel hypothesis that men and women with CKD possess different risk factors. The objective of this study was to develop a risk prediction model among gender by using seven machine learning algorithms to predict early CKD.

## 2. Materials

### 2.1. Participants

In this study, we collected 19,270 valid health screening records from 32 health screening clinics and three specialized laboratories; this included 19,270 screening results that were recorded between 1 January 2015 and 31 December 2019. Of the 8073 male and 11,197 female records, 5101 had CKD patients and 14,169 healthy samples. The average age of CKD patients was 69.19, in the range of 58.45–79.93 years.

### 2.2. Instruments

Seven machine learning techniques were used to predict early CKD, including support vector machine (SVM), linear discriminant analysis (LDA), logistic regression (LGR), C4.5 decision tree (C4.5), classification and regression tree (CART), random forest (RF), and C5.0 decision tree (C5. 0) [15,16,17].

CART is a decision tree algorithm that uses a binary process to sequentially divide the data space and generate a simple prediction rule in each partition. CART not only solves classification problems but also performs a regression analysis, which means that this method generates a classification tree when the target of the predictor variable is clear. The first step in the CART analysis is to construct all decision rules through a binary decomposition process; the second step is to prune the overgrown trees to eliminate unnecessary rule trees; and the last step is to determine the best tree rules using cross-validation.

C4.5 is a decision tree algorithm that uses the ID3 algorithm to produce an improved iterative binary tree. It selects the attributes of the decision tree at each node according to the information acquisition rate. According to information theory and probability statistics, entropy is one of several ways to measure the average level of how many different types exist in a dataset. C4.5 reflects the training set and gradually develops a tree structure for attribute variables according to the maximum information gain ratio.

LDA is a widely used method for dimensionality reduction and classification. For the data to be analyzed, LDA looks for different categories with their own dimensional space in which the distance between samples of different categories gradually increases, and, contrarily, the distance between samples of the same category decreases. In the learning process, LDA can obtain the function to project different types of samples into low-dimensional space and then performs feature decomposition to calculate the best projection.

The C5.0 algorithm can generate decision trees or rule sets. The C5.0 model splits the samples based on the maximum information gain. The sample subset determined by the first split is then split again, usually based on another field, and this process is repeated to cause the sample subset to split until it cannot be split any further. Finally, the lowest-level split is re-captured, and the sample subsets that do not significantly contribute to the model value are pruned.

RF is an overall classification method based on the statistical learning theory that combines several separate classification trees. RF is a supervised machine learning algorithm that considers the unweighted majority of classified votes. It first uses bootstrapping to select various random variables as the training dataset. This widely used algorithm uses random sampling and replacement to simultaneously reduce variance and avoid overfitting. The classification tree of the selected sample is then used to construct the training process, using a large number of classification trees to form the RF from selected samples. CART is a classification method that is widely used for RF modeling. Finally, all classification trees are merged, with voting for each category, and then the winning category is selected according to the number of votes to obtain the final classification result.

SVM is a machine learning algorithm that is based on the principle of structural risk minimization and is used to estimate the function by minimizing the upper limit of the generalization error. SVM modeling can initially use a linear or nonlinear kernel function to map the input vector to a feature space. Then, in the feature space, SVM tries to find the best linear division to construct a hyperplane that separates various types.

LGR is a classification algorithm rather than a regression algorithm. Usually, the known independent variable is used to predict the value of a discrete dependent variable (for example, a binary value of 0/1, yes/no, and true/false). It predicts the probability of an event by fitting a logit function, thus producing a probability value between 0 and 1.

### 2.3. Procedure and Data Analysis

Previous studies revealed that gender is associated with CKD deterioration [3,22,23]. The urine protein (PRO)-to-urine creatinine ratio (UPCR) is the most important risk factor this consequence [24,25,26]. The UPCR is used for the next section of analyzing the impact of gender differences to predict the risk factors for CKD. Important risk factors of CKD were analyzed through discussions with clinical experts and a review of the relevant literature. The data were cleaned, re-encoded, ranked in importance using the gain ratio and information gain classifier, and then divided into training and test sets (at a 7:3 ratio) through 10 rounds of random sampling. Gender prediction and a clinical risk factor analysis were then performed using seven machine learning classifiers. On the basis of expert input and the relevant literature, 11 independent predictive variables were selected (see Table 2).

Regarding the imbalance issue, we adopted the following kinds of sampling techniques: manual extraction technique, synthetic minority over-sampling technique (SMOTE), and SpreadSubsample. Manual extraction is an easy extraction technique, and it under-sampled the majority category data at random. The synthetic minority over-sampling technique (SMOTE) is an over-sampled minority category data that creates synthetic samples, and it makes extra data by actual examples. SpreadSubsample is one of under sampling method to solve the issue of data imbalance, and it can be balanced with the minority class. Three types of extraction techniques were used to solve the majority and minority data imbalance problem. Using the above classification methods, we used the rpart package (version R4.1.15) to build the CART prediction model. The OptimClassifier suite (version R0.1.5) was used to determine the tree depth, the number of observations in the terminal node, and the pruning tree parameters to search for the best parameter set to generate the CART model [22]. The RWeka suite (version R0.4–42) was used to construct the C4.5 model. The caret suite (version R 6.0–84) was used to identify the best parameter set to effectively build the C4.5 model. LDA used the MASS suite (version R7.3–51.5). The ELM model was constructed by running the elmNN package in the R1.0 version [23]. The caret suite (version R6.0–84) was used to adjust important hyperparameters to find the best number of hidden layer neurons to generate the best ELM model [23]. Classification accuracy was assessed based on accuracy, sensitivity, specificity, and the receiver operating characteristic curve by estimating the area under the curve (AUC).

## 3. Results

### 3.1. Risk Factors for Predicting CKD

This study used the adult health screening data to predict risk factors for CKD. The results of the Chi-square statistical test are summarized in Table 3. Gender, age, urinary red blood cells (RBC), fasting blood glucose (GLU), neutral lipid (triglyceride) (TG), high-density lipoprotein (HDL), PRO, and the urinary protein-to-creatinine ratio (UPCR) were all found to be statistically significant. In terms of gender, the majority of male participants were found to have CKD (48.3% CKD vs. 39.6% healthy). The proportion of participants with abnormal RBC in the CKD group was higher than that in the healthy participants group (23.2% vs. 19.1%). The GLU in the CKD group was higher than that in the healthy participants group (20.7% vs. 18.8%). Compared with the healthy group, the CKD group had a higher proportion of TG abnormalities (60.6% vs. 58.5%). In the CKD group, the proportion of PRO abnormalities was higher than that in the healthy participants group (82.1% vs. 35.0%), as was the UPCR (67.9% vs. 12.7%). No significant differences were found for serum total cholesterol (*p* = 0.491), low-density lipoprotein (*p* = 0.782), or albumin (*p* = 0.457).

### 3.2. Prediction Models for CKD

A comparison among the three modes of extraction techniques (Manual, SMOTE, and SpreadSubsample) was performed by using seven machine learning techniques to train the predictive risk models of each mode for male and female groups, as shown in Figure 1 and Figure 2. On the basis of manual extraction techniques, risk factors in the male participants group were age, RBC, GLU, HDL, PRO, and UPCR, with LGR exhibiting the highest AUC (0.834) (Figure 1a), whereas in the female participants group, they were age, RBC, T-CHO, PRO, and UPCR, with LDA exhibiting the highest AUC (0.8485) (Figure 2a). Common risk factors for both genders were age, RBC, PRO, and the UPCR. When using the synthetic minority oversampling technique (SMOTE) extraction analysis, the most significant risk factors in the male participants group were age, RBC, GLU, HDL, PRO, and the UPCR, with LGR exhibiting the highest AUC (0.824) (Figure 1b), whereas in the female participants group, they were age, RBC, T-CHO, HDL, PRO, and the UPCR, with LGR exhibiting the highest AUC (0.837) (Figure 2b). The common risk factors for both genders were age, RBC, HDL, PRO, and the UPCR. When performing SpreadSubsample extraction analysis techniques, the most significant risk factors in the male participants group were age, RBC, GLU, HDL, PRO, and the UPCR, with LGR exhibiting the highest AUC (0.833) (Figure 1c), whereas in the female participants group, they were age, RBC, T-CHO, PRO, and the UPCR, with LGR exhibiting the highest AUC (0.8472) (Figure 2c). The common statistically significant risk factors for both genders were age, RBC, PRO, and the UPCR. Regardless of the extraction method, the top three risk factors for deteriorating CKD in male participants were the UPCR, PRO, and age, whereas in females, they were the UPCR, PRO, and age.

The UPCR was further used to divide the following stage, which was used to analyze gender differences and predict CKD risk factors based on UPCR status. Different sampling method results are presented below.

Through manual extraction analysis techniques, the most significant risk factors for a normal UPCR in male participants were age, RBC, TG, HDL, and PRO, with LGR exhibiting the highest AUC (0.715), whereas in female participants, they were age, RBC, GLU, TG, HDL, and PRO, with LGR exhibiting the highest AUC (0.7039). Common risk factors for both genders were age, RBC, TG, HDL, and PRO. Using SMOTE extraction analysis, the most significant risk factors for a normal UPCR in male participants were age, RBC, TG, T-CHO, HDL, and PRO, with LDA exhibiting the highest AUC (0.693), whereas in females, they were age, RBC, GLU, TG, T-CHO, HDL, LDL, and PRO, with LDA exhibiting the highest AUC (0.7061). Common risk factors for UPCR in both genders were age, RBC, TG, T-CHO, HDL, and PRO. Using SpreadSubsample extraction analysis techniques, the most significant risk factors of a normal UPCR in male participants were age, RBC, TG, and PRO, with LDA exhibiting the highest AUC (0.657), whereas in females, they were age, RBC, GLU, TG, and PRO, with LDA exhibiting the highest AUC (0.7142). Common statistically significant risk factors for UPCR in both genders were age, RBC, TG, and PRO. Regardless of the extraction method, the top three risk factors for CKD deterioration under normal UPCR in male participants were PRO, RBC, and age, whereas for females, they were PRO, age, and RBC.

The most significant risk factors for abnormal UPCR in male participants included age, RBC, GLU, HDL, and PRO, with LDA exhibiting the highest AUC (0.741), whereas in females, they were age, RBC, GLU, GLU, T-CHO, and PRO, with LGR exhibiting the highest AUC (0.7318). Common risk factors for UPCR abnormalities in both genders were age, RBC, GLU, and PRO.

### 3.3. Decision Tree Analysis

This study sought to set up a precise prediction from routine health screenings of asymptomatic adult patients. A comprehensive clinical prevention method that considers all these factors is needed to successfully solve the problem of high-risk exposure in the adult population and aid in successful prevention specifically for these populations. As shown in Figure 3, through decision tree analyses, all samples pass through 12 subsets of different branches, from the root node to the leaf node, by conditional screening. As mentioned earlier, gender has a strong influence on CKD interpretation and was selected as the root node of the classification decision tree. The second-order decision tree node is the UPCR. The third-order decision tree is generated by age, RBC, TG, PRO, T-CHO, and LDL, with classification prediction accuracy ranging from 55.2% to 84.9%. Table 4 presents the results of the 12 combinations of conditions.

## 4. Discussion

Effective prediction of patient risk of developing CKD allows for early detection and treatment. The data analysis results show that the most important risk factor in CKD prediction in both genders was the UPCR, followed by PRO and age. Regardless of a normal or abnormal UPCR in male participants, the top three risk factors for CKD were PRO, age, and RBC, and the same results were observed in the females analyzed for UPCR.

Our findings are consistent with previous findings in that it was demonstrated that gender is very important in predicting CKD [9,10,11], and also in accordance with the National Health Research Institutes Annual Report on Kidney Disease regarding the UPCR [3] and the RBC [24]. The findings of the ALB and GLU are consistent with our previous studies [18]. Similarly, Kshirsagar et al. [25] reported that age, and gender are important risk factors for prediction of chronic kidney disease. Other risk factors included UPC, age, RBC, GLU, TG, T-CHO, HDL-C, LDL-C, ALB, and PRO were important risk factors in this gender difference analysis [27]. The result is consistent with previous reports, including research reports on the role of UPCR and RBC in CKD from the National Institutes of Health [3]. The results for PRO are consistent with [28,29,30], and the findings for ALB and GLU are consistent with previous studies [31,32,33,34]. Other studies found that the prevalence of CKD is often positively correlated with diabetes, hypertension, and obesity [35,36,37], which is also consistent with the results of the present study.

Nevertheless, the key analyzed risk factor in this study was gender. This study indicated that CKD epidemiology differs by gender, affecting more females than males. Although females have a higher prevalence of chronic kidney disease, men with CKD have a faster progression to kidney failure and represent a greater proportion of the population with kidney failure. On the other hand, females are less likely to receive dialysis on observation. A possible alternative explanation for the existing gender differences in CKD outcomes is that females tend towards conservative care. More investigation is still needed to identify biologic and psychosocial factors underlying these gender disparities [38,39]. In this light, machine learning-based risk prediction models can provide supportive evidence for the robustness of early clinical risk assessments and CKD prediction.

## 5. Conclusions

Previous research has highlighted the need for new technologies to diagnose, prevent, treat, and raise awareness about CKD. The specific and autonomous predictive model of CKD could improve clinical care. As a result, the application of AI has great potential in the field of nephrology. Assessing kidney health in different settings offers a more precise, effective, and often more practical method. This study used a large dataset of adult health screening results to identify important risk factors for gender differences in predicting CKD. The results indicate that, regardless of gender, health screenings should emphasize the UPCR, PRO, and age. For male UPCRs, attention should be paid to PRO, age, and urine RBC. For female UPCRs, attention should be paid to PRO, age, and urine RBC. In addition, this research also offers several predictive models that, unlike traditional statistical methods, are effective in predicting a patient’s risk of developing CKD for early detection and treatment.

Although there is much evidence that artificial intelligence appears to be a potentially useful tool with which to overcome the challenges faced by nephrologists. The development of this approach in nephrology is still limited by the huge heterogeneity (not always readily available) of clinical data that needs to be integrated to optimize the performance of these models. Further investigation of prediction needs to be conducted after CKD diagnosis, and follow-up is necessary to track patients. Additionally, previous studies often showed imbalances in data categories; we recommend that these experiments be trained in a broad range of cases that represent the entire population, representing the current large-scale implementation and limitations of individual patient treatment. In summary, the aging of the population and the chronic nature of several diseases seem to justify the prevalence of CKD and the increase in acute renal insufficiency. New technologies, including artificial intelligence, are a valuable and potentially useful tool that can optimize work in this area and improve the management and treatment of patients with kidney disease.

## Figures and Tables

**Figure 1 ijerph-19-01219-f001:**
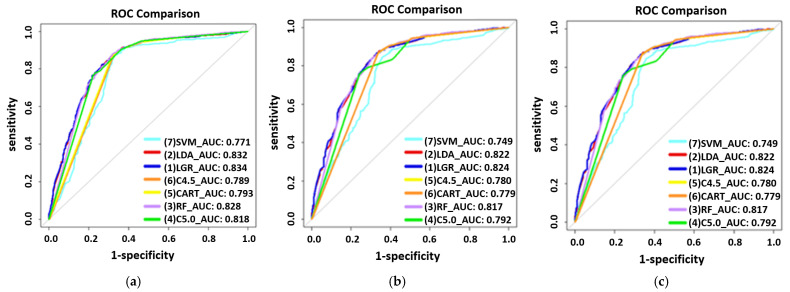
The receiver operating characteristic (ROC) curves for comparing extraction methods for males. (**a**) Manual extraction. (**b**) SMOTE extraction. (**c**) SpreadSubsample extraction.

**Figure 2 ijerph-19-01219-f002:**
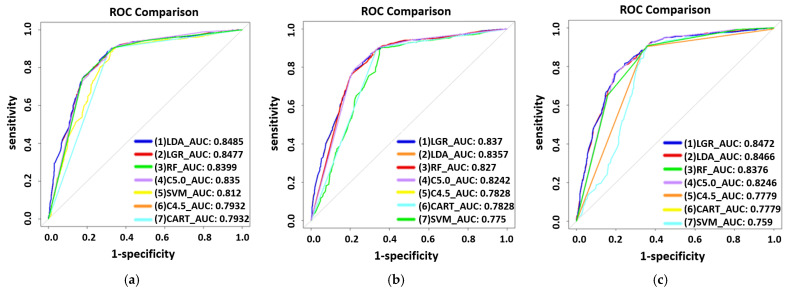
The receiver operating characteristic (ROC) curves for comparing extraction methods for females. (**a**) Manual extraction. (**b**) SMOTE extraction. (**c**) SpreadSubsample extraction.

**Figure 3 ijerph-19-01219-f003:**
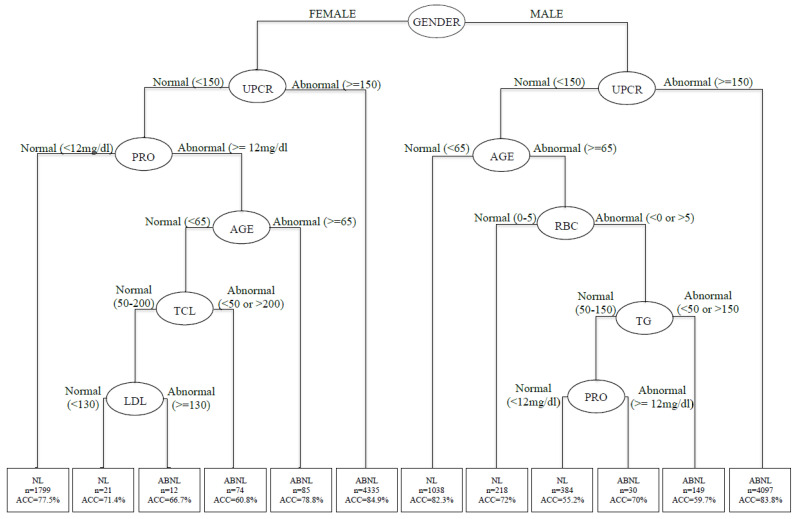
Decision tree for predicting variables by gender. RBC: red blood cell; GLU: glucose; TG: triglycerides; T-CHO: total cholesterol; HDL: high-density cholesterol; LDL: low-density cholesterol; ALB: albumin; PRO: proteinuria; UPCR: urine protein-to-creatinine ratio; eGFR: estimated glomerular filtration rate.

**Table 1 ijerph-19-01219-t001:** Five stages of chronic kidney disease.

Stages	Description	GFR Value
1	CKD with normal or high GFR	≥90 mL/min/1.73 m^2^
2	Mild CKD	60–89.9 mL/min/1.73 m^2^
3	Moderate CKD	30–59.9 mL/min/1.73 m^2^
	3a	45–59.9 mL/min/1.73 m^2^
	3b	30–44.9 mL/min/1.73 m^2^
4	Severe CKD	15–29.9 mL/min/1.73 m^2^
5	End stage CKD	<15 mL/min/1.73 m^2^

GFR: glomerular filtration rate; 3a: stage 3a of kidney disease; 3b: stage 3b of kidney disease.

**Table 2 ijerph-19-01219-t002:** Data sources and variable codes.

Variables	Name	Normal Range
X1	Gender	1 male/2 female
X2	Age	Continuous
X3	RBC	0–5
X4	GLU	70–100
X5	TG	50–150
X6	T-CHO	50–200
X7	HDL	>40
X8	LDL	<130
X9	ALB	3.5–5.0
X10	PRO	Random < 12 mg/dL
X11	UPCR	<150
Y	eGFR	1. <90 mL/min/1.73
		2. ≥90 mL/min/1.73 m^2^

RBC: red blood cell; GLU: glucose; TG: triglycerides; T-CHO: total cholesterol; HDL: high-density cholesterol; LDL: low-density cholesterol; ALB: albumin; PRO: proteinuria; UPCR: urine protein-to-creatinine ratio; eGFR: estimated glomerular filtration rate.

**Table 3 ijerph-19-01219-t003:** Descriptive analysis of variables.

Items	Healthy	CKD	*p*-Value	χ^2^
*n* (%)	14,169 (73.5%)	5101 (26.5%)		
Gender				
Male	5608 (39.6%)	2465 (48.3%)	<0.001 **	117.817
Female	8561 (60.4%)	2636 (51.7%)		
Age				
Mean (±SD)	63.37 ± 11.56	69.19 ± 10.74	<0.001 *	699.271
RBC				
Normal	11,460 (80.9%)	3917 (76.8%)	<0.001 **	38.956
Abnormal	2709 (19.1%)	1184 (23.2%)		
GLU				
Normal	2667 (18.8%)	1055 (20.7%)	0.004 **	8.321
Abnormal	11,502 (81.2%)	4046 (79.3%)		
TG				
Normal	5878 (41.5%)	2012 (39.4%)	0.011 *	6.466
Abnormal	8291 (58.5%)	3089 (60.6%)		
T-CHO				
Normal	9198 (64.9%)	3284 (64.4%)	0.491	0.474
Abnormal	4971 (35.1%)	1817 (35.6%)		
HDL				
Normal	11,954 (84.4%)	4369 (85.6%)	0.029 *	4.763
Abnormal	2215 (15.6%)	732 (14.4%)		
HDL				
Normal	11,400 (80.5%)	4095 (80.3%)	0.782	0.076
Abnormal	2769 (19.5%)	1006 (19.7%)		
ALB				
Normal	14,162 (100.0%)	5097 (99.9%)	0.457	0.553
Abnormal	7 (0.0%)	4 (0.1%)		
PRO				
Normal	9203 (65.0%)	915 (17.9%)	<0.001 *	3324.451
Abnormal	4966 (35.0%)	4186(82.1%)		
UPCR				
Normal	12,364 (87.3%)	1639 (32.1%)	<0.001 *	5739.411
Abnormal	1805 (12.7%)	3462 (67.9%)		

** *p*-value < 0.01; * *p*-value < 0.05. RBC: red blood cell; GLU: glucose; TG: triglycerides; T-CHO: total cholesterol; HDL: high-density cholesterol; LDL: low-density cholesterol; ALB: albumin; PRO: proteinuria; UPCR: urine protein-to-creatinine ratio.

**Table 4 ijerph-19-01219-t004:** Classification and regression tree (CART)decision rule for predicting variables by gender.

Rule No.	The Composition of Risk Factors	No.	Status	Accuracy
1	Gender (Female) + UPCR (<150) + PRO (<12)	1799	Non-CKD	77.5%
2	Gender (Female) + UPCR (<150) + PRO (≥12) + Age (<65) + T-CHO (50–200) + LDL (<130)	21	Non-CKD	71.4%
3	Gender (Female) + UPCR (<150) + PRO (≥12) + Age (<65) + T-CHO (50–200) + LDL (≥130)	12	CKD	66.7%
4	Gender (Female) + UPCR (<150) + PRO (≥12) + Age (<65) + T-CHO (<50 or >200)	74	CKD	60.8%
5	Gender (Female) + UPCR (<150) + PRO (≥12) + Age (≥65)	85	CKD	78.8%
6	Gender (Female) + UPCR (≥150)	4335	CKD	84.9%
7	Gender (Male) + UPCR (<150) + Age (<65)	1038	Non-CKD	82.3%
8	Gender (Male) + UPCR (<150) + Age (≥65) + RBC (0–5)	218	Non-CKD	72%
9	Gender (Male) + UPCR (<150) + Age (≥65) + RBC (<0 or >5) + TG (50–150) + PRO (<12)	384	Non-CKD	55.2%
10	Gender (Male) + UPCR (<150) + Age (≥65) + RBC (<0 or >5) + TG (50–150) + PRO (≥12)	30	CKD	70%
11	Gender (Male) + UPCR (<150) + Age (≥ 65) + RBC (<0 or >5) + TG (<50 or >200)	149	CKD	59.7%
12	Gender (Male) + UPCR (≥ 150)	4097	CKD	83.8%

RBC: red blood cell; TG: triglycerides; T-CHO: total cholesterol; LDL: low-density cholesterol; PRO: proteinuria; UPCR: urine protein-to-creatinine ratio.

## Data Availability

Data are available from the Institutional Review Board of Chung Shan Medical University Hospital for researchers who meet the criteria for access to confidential data. Requests for the data may be sent to the Chung Shan Medical University Hospital Institutional Review Board, Taichung City, Taiwan (e-mail: irb@csh.org.tw).

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
