# Peer review of "Associations between Sex and Risk Factors for Predicting Chronic Kidney Disease"

_ijerph, 2022, doi:10.3390/ijerph19031219_

Round 1

Reviewer 1 Report

I would like to start by congratulating the authors and congratulating them for having decided to investigate an area where there is still so much to discover, but also for having decided to share this protocol with the rest of the scientific community, so that science can evolve.

This is an article about Associations between Sex and Risk Factors for Predicting Chronic Kidney Disease.

All comments, questions and suggestions presented are constructive and try to improve the article, after several careful readings.

Abstract

The conclusion is not clear and unambiguous. After all, what is the conclusion of this study?

Keywords

Repetitions with expressions that are in the title should be avoided. Whenever possible, keywords should be Mesh.

Introduction

In the caption of Table 1 the reference must be placed, despite being in the text.

What is the purpose of this study? The same must be defined in the abstract.

Results

Is it possible to get more quality of the graphics?

General comments

Very interesting article, well written and with clear potential to be published.

Authors must include captions in tables and figures, whenever they have abbreviations.

This article can become somewhat confusing to read exactly because of the amount of abbreviations it presents. Authors should make reading easier.

This article has the possibility of changing the clinical practice of nephrologists all over the world and of changing the way in which clinical decisions are made. But for this to happen, the authors have to try to facilitate the reading of those who are going to read this article.

The article as presented must submitted to major revision.

Author Response

Response to Reviewer 1 Comments

Q: I would like to start by congratulating the authors and congratulating them for having decided to investigate an area where there is still so much to discover, but also for having decided to share this protocol with the rest of the scientific community, so that science can evolve.

This is an article about Associations between Sex and Risk Factors for Predicting Chronic Kidney Disease.

All comments, questions and suggestions presented are constructive and try to improve the article, after several careful readings.

Q1: Abstract

The conclusion is not clear and unambiguous. After all, what is the conclusion of this study?

Response:

Thanks for suggestion to improve our work. To address the concerns, additional sentences have been appended as follows.

“As a result, this study proposed a prediction model with acceptable prediction accuracy. The model supports doctors in diagnosis and treatment and achieves the goal of early detection and treatment. Based on the academic foundation of evidence-based medicine, this research through practical perspectives and develops an CKD risk prediction model based on machine learning. The model has proven to support as much as possible the prediction of early clinical risk of CKD to improve the efficacy and quality of clinical decision-making.”

Q2: Keywords

Repetitions with expressions that are in the title should be avoided. Whenever possible, keywords should be Mesh.

Response:

We thank the reviewer for reminding us this suggestion. We have avoided repetition of the expression in the title and keywords are matched Mesh Heading or Entry Term.

Q3: Introduction

In the caption of Table 1 the reference must be placed, despite being in the text.

Thanks for your suggestions. The abbreviations of Table 1 have been added in the revised manuscript.

Q4: Introduction

What is the purpose of this study? The same must be defined in the abstract.

Response:

We had revised the sentence at line 23 and 24.

“The purpose of this study was to examine whether gender differences affect the risk factors of early CKD prediction.”

Q5: Results

Is it possible to get more quality of the graphics?

Response:

We improved the resolution of the figures. Thank you for your advice.

Q6: General comments

Very interesting article, well written and with clear potential to be published. Authors must include captions in tables and figures, whenever they have abbreviations.  

Response:

Thanks for your suggestions. The abbreviations of all tables and figures have been added in the revised manuscript.

Q7: This article can become somewhat confusing to read exactly because of the amount of abbreviations it presents. Authors should make reading easier. This article has the possibility of changing the clinical practice of nephrologists all over the world and of changing the way in which clinical decisions are made. But for this to happen, the authors have to try to facilitate the reading of those who are going to read this article. The article as presented must submitted to major revision.

Response:

The authors are very thankful for the given opportunity for amend the clinical issues as a great reflect of your interest in the manuscript. Has been re-written

at line 53-55, 60-61, 67, 84-86, 88, 94-97, 100, 103-105, 108-113, 126-129, 150, 287, 300-301, 309, 322-323, 326, 343. The detailed description has been edited in the revised version.

Reviewer 2 Report

Dear authors, in my review of your work I indicated the following aspects to be revised, although I only noticed changes in the bibliographic citations in the correct format of the journal and the change in figure 3, before figure 5.

  1. Regarding the introduction, when the study addresses the use of prediction techniques, recent information on the state of the art in this respect should be included. In other words, studies on the use of prediction techniques used by the authors with respect to the problem studied. Likewise, the authors should generally include a greater number of bibliographical references to support their work, as only 34 citations are found in the bibliography, of which only 41.17% are citations updated in the last five years. As this is such a new topic, I believe that there should be more up-to-date studies in the scientific literature.
  2. At the end of the introduction section, the authors should clearly describe their research objectives and hypotheses.
  3. The method section is not adequately structured as this section should contain the following sub-sections: participants (it should include a description of the sample and the sampling applied for its selection as well as information on informed consent and the number of the positive report from the ethics committee of the institution where the study was conducted), instruments (all instruments used to find the values of the study variables as well as their reliability indicators should be clearly described), procedure (a clear step-by-step description of how the study was carried out should be provided; the authors describe this information in several sections, although it is not ordered under this heading), data analysis (the relationship between the research questions and the tests or algorithms to be used to test them should be indicated here).
  4. The results section should report one by one the results found for each of the hypotheses or research questions proposed.
  5. In the conclusions section, the results must be related to the results of the research that has served as a reference for the study, including the numbers in each of the concluding sentences. In this section, opinions that are not refuted either by the findings of other research or by the novelty found with respect to them cannot be included. In addition, the authors should refer to the limitations of the study and future lines of research.
  6. References: as indicated above, the number of references should be increased and updated in the last 5 years. Also, the authors should adjust the references to the journal's rules for citing.

Author Response

Response to Reviewer 2 Comments

Q1: Dear authors, in my review of your work I indicated the following aspects to be revised, although I only noticed changes in the bibliographic citations in the correct format of the journal and the change in figure 3, before figure 5. Regarding the introduction, when the study addresses the use of prediction techniques, recent information on the state of the art in this respect should be included. In other words, studies on the use of prediction techniques used by the authors with respect to the problem studied. Likewise, the authors should generally include a greater number of bibliographical references to support their work, as only 34 citations are found in the bibliography, of which only 41.17% are citations updated in the last five years. As this is such a new topic, I believe that there should be more up-to-date studies in the scientific literature.

Response:

We have added several important references to support our work, details are as follows

  • Shih, C-C; Chen, S-H; Chen, G-D; Chang, C-C; Shih, Y-L. Development of a Longitudinal Diagnosis and Prognosis in Patients with Chronic Kidney Disease: Intelligent Clinical Decision-making Scheme. Int J Environ Res Public Health. 2021, 18, 12807.
  • Chang, C-C; Huang, T-H; Shueng, P-W; Chen, S-H; Chen, C-C; Lu, C-J; Tseng, Y-J. Developing a Stacked Ensemble-based Classification Scheme to Predict Second Primary Cancers in Head and Neck Cancer Survivors. Int J Environ Res Public Health. 2021, 18, 12499.
  • Chang, C-C; Chen, C-C; Cheewakriangkrai, C; Chen, Y-C; Yang, S-F. Risk Prediction of Second Primary Endometrial Cancer in Obese Women: A Hospital-Based Cancer Registry Study. Int J Environ Res Public Health. 2021, 18, 8997.
  • Chan, C-L; Chang, C-C. Big Data, Decision Models, and Public Health. Int J Environ Res Public Health. 2020, 17, 6723.

Q2: At the end of the introduction section, the authors should clearly describe their research objectives and hypotheses.

Response:

Thanks for your comment. The Introduction Section is rephrased to incorporate the reviewer’s suggestions. The following description has been edited in the revised version.

 “The present study sought to evaluate the novel hypothesis that men and women with CKD possess different risk factors. The objective of this study is to develop a risk prediction model among gender by using seven machine-learning algorithms to predict early CKD.”

Q3: The method section is not adequately structured as this section should contain the following sub-sections: participants (it should include a description of the sample and the sampling applied for its selection as well as information on informed consent and the number of the positive report from the ethics committee of the institution where the study was conducted), instruments (all instruments used to find the values of the study variables as well as their reliability indicators should be clearly described), procedure (a clear step-by-step description of how the study was carried out should be provided; the authors describe this information in several sections, although it is not ordered under this heading), data analysis (the relationship between the research questions and the tests or algorithms to be used to test them should be indicated here).

Response:

The Material Section is revised and added two sub sessions, 2.1 Data Source and 2.2 Method. The detailed description has been edited in the revised version.

Q4: The results section should report one by one the results found for each of the hypotheses or research questions proposed.

Response:

We thank the reviewer for reminding us this suggestion.

We have added some references and detailed description for research questions in the Result section.

  • National Health Research Institutes Annual Report on Kidney Disease in Taiwan. Available online: http://w3.nhri.org.tw/nhri_org/rl/lib/NewWeb/nhri/ebook/39000000448683.pdf (accessed on 27 Dec 2021).
  • The National Health Insurance Statistics. 2017. Available online: https://www.nhi.gov.tw/english/Content_List.aspx?n=0D39BCF70F478274&topn=616B97F8DF2C3614 (accessed on 27 Dec 2021).
  • Shih, C-C; Chen, S-H; Chen, G-D; Chang, C-C; Shih, Y-L. Development of a Longitudinal Diagnosis and Prognosis in Patients with Chronic Kidney Disease: Intelligent Clinical Decision-making Scheme. Int J Environ Res Public Health. 2021, 18, 12807.
  • Korbut, A.I.; Klimontov, V.V.; Vinogradov, I.V.; Romanov, V.V. Risk factors and urinary biomarkers of non-albuminuric and albuminuric chronic kidney disease in patients with type 2 diabetes. World J. Diabetes 2019, 10, 517–533.

Q5: In the conclusions section, the results must be related to the results of the research that has served as a reference for the study, including the numbers in each of the concluding sentences. In this section, opinions that are not refuted either by the findings of other research or by the novelty found with respect to them cannot be included. In addition, the authors should refer to the limitations of the study and future lines of research.

Response:

Thank you for your valuable comment.

It has been included further research directions in Conclusion section.

Q6: References: as indicated above, the number of references should be increased and updated in the last 5 years. Also, the authors should adjust the references to the journal's rules for citing.

Response:

We have added several important references to support our work, details are as follows

  • Shih, C-C; Chen, S-H; Chen, G-D; Chang, C-C; Shih, Y-L. Development of a Longitudinal Diagnosis and Prognosis in Patients with Chronic Kidney Disease: Intelligent Clinical Decision-making Scheme. Int J Environ Res Public Health. 2021, 18, 12807.
  • Chang, C-C; Huang, T-H; Shueng, P-W; Chen, S-H; Chen, C-C; Lu, C-J; Tseng, Y-J. Developing a Stacked Ensemble-based Classification Scheme to Predict Second Primary Cancers in Head and Neck Cancer Survivors. Int J Environ Res Public Health. 2021, 18, 12499.
  • Chang, C-C; Chen, C-C; Cheewakriangkrai, C; Chen, Y-C; Yang, S-F. Risk Prediction of Second Primary Endometrial Cancer in Obese Women: A Hospital-Based Cancer Registry Study. Int J Environ Res Public Health. 2021, 18, 8997.
  • Chan, C-L; Chang, C-C. Big Data, Decision Models, and Public Health. Int J Environ Res Public Health. 2020, 17, 6723.

Reviewer 3 Report

The paper presents insights of the application of machine-learning techniques on decision tree elaboration for chronic kidney disease risk factors.

I suggest modifications in the materials-methods section. 

The number of "indipendent" risk factors can be abbreviated as proteinuria and urine protein/creatinine ratio are directly related. In clinical practice urine protein/creatinine ratio presents a higher precision than proteinuria (expressed as protein concentration per volume).

A description of the patient population (age, hypertension, diabetes, smoking, CKD classes of the patients screened) is missing.

The term "manual extraction techinques" should be mentioned in the Materials section.

A description of the "synthetic minority oversampling technique" and of the "SpreadSubsample extraction analysis technique" are missing in the Materials section.

Author Response

Response to Reviewer 3 Comments

The paper presents insights of the application of machine-learning techniques on decision tree elaboration for chronic kidney disease risk factors.

Q1: I suggest modifications in the materials-methods section. 

The number of "indipendent" risk factors can be abbreviated as proteinuria and urine protein/creatinine ratio are directly related. In clinical practice urine protein/creatinine ratio presents a higher precision than proteinuria (expressed as protein concentration per volume).

Response:

Thanks for suggestion to improve our work. The Material Section is revised and added two sub sessions, 2.1 Data Source and 2.2 Method. The detailed description has been edited in the revised version.

Q2: A description of the patient population (age, hypertension, diabetes, smoking, CKD classes of the patients screened) is missing.

Response:

Thanks for suggestion to improve our work. we add the following descriptions in the result section of revised version.

“Of the 8073 male and 11197 female records, 5101 had CKD patients and 14169 healthy samples. The average age of CKD patients was 69.19, range 58.45-79.93 years, shows in table 3….”

Q3: The term "manual extraction techinques" should be mentioned in the Materials section.

Response:

Thanks for your suggestion. The introduction of manual extraction technique is included in the material section of revised manuscript.

Q4: A description of the "synthetic minority oversampling technique" and of the "SpreadSubsample extraction analysis technique" are missing in the Materials section.

Response:

Thanks for your suggestion. The introduction of each method is included in the material section of revised manuscript.

Round 2

Reviewer 1 Report

Authors made major revision as addresed. I'm satisfied with the final article.

Author Response

Response: Thank you.

Reviewer 2 Report

Relation of the first revision to the changes made by the authors in the new version of the manuscript

First review:

  1. With regard to the introduction, the study should include recent information on the state of the art in this respect. In other words, studies on the use of prediction techniques used by the authors with respect to the problem studied. Likewise, the authors should include, in general, a greater number of bibliographical references to support their work, as only 34 citations are found in the bibliography, of which only 41.17% are citations updated in the last five years. As this is such a new topic, I believe that there should be more up-to-date studies in the scientific literature.

Second review

The authors have included new paragraphs in the introduction and new references in the bibliography, although they have not included the numbers of the new references in the introduction.

First revision:

  1. At the end of the introductory section, the authors should clearly describe their research objectives and hypotheses.

Second review

The authors have clearly included the hypotheses of the study.

First review:

  1. The method section is not adequately structured as this section should contain the following sub-sections: participants (should include a description of the sample and the sampling applied for its selection as well as information on informed consent and the number of the positive report from the ethics committee of the institution where the study was conducted), instruments (all instruments used to find the values of the study variables as well as their reliability indicators should be clearly described), procedure (a clear step-by-step description of how the study was carried out should be provided; the authors describe this information in several sections, although it is not ordered under this heading), data analysis (the relationship between the research questions and the tests or algorithms to be used to test them should be indicated here).

Second revision

This section is still not correctly structured, the authors offer sample data but include them in the materials section in the Data Source sub-section, in my opinion this is not correct, this data should be included in the participants section and this should be in the method section. On the other hand, the method section cannot be a sub-section of the materials section, method itself is a section that should include the following sub-sections (participants, instruments, procedure and data analysis). Therefore, these aspects should be reformulated.

First revision:

  1. The results section should relate one by one the results found in each of the hypotheses or research questions proposed.

Second revision

This aspect is still not adequately addressed.

First review:

  1. In the conclusions section, the results must be related to the results of the research that has served as a reference for the study, including the numbers in each of the concluding sentences. In this section, opinions that are not refuted either by the findings of other research or by the novelty found with respect to them cannot be included. In addition, the authors should refer to the limitations of the study and future lines of research.

Second review

I do not see that they have addressed the section on conclusions, it should be included with the recommendations given in the first review.

First review:

  1. References, as already indicated, the number of references should be increased and updated in the last 5 years. Also, the authors should adjust the references to the journal's rules for citing.

Second revision

The authors have introduced 8 new references, of which updated in the last five years are 7.

First revision:

  1. Figure 5 needs to be improved in terms of presentation.

Second revision

There is no Figure 5 in this new version.

First revision:

  1. On the other hand, following the journal's template, authors are encouraged to take it as a reference, all the information is available at this link https://www.mdpi.com/journal/ijerph/instructions , the following sections are missing: Author Contributions, Funding, Institutional Review Board Statement, Informed Consent Statement, Data Availability Statement, Acknowledgments, and Conflicts of Interest.

Second revision

The authors have included the sections Author Contributions, Funding and Conflicts of Interest.

 However, the following sections are missing: Institutional Review Board Statement, Informed Consent Statement, Data Availability Statement, Acknowledgments, and Conflicts of Interest.

  1. The discussion section cannot include figures or tables (these data should be included in the results section), this section should address the results found, the discrepancies with previous research, the limitations of the study and future lines of research continuity. It should therefore be redrafted in its entirety.

Author Response

We would like to thank the reviewer for his/her positive and very thorough comments on the manuscript. According to reviewers' advices, we have incorporated as many of the reviewer's suggestions as possible into the revised manuscript. Also, the paper is much improved as a result and will enhance readability of this article. 
